# Microfluidic Liquid Biopsy Minimally Invasive Cancer Diagnosis by Nano-Plasmonic Label-Free Detection of Extracellular Vesicles: Review

**DOI:** 10.3390/ijms26136352

**Published:** 2025-07-01

**Authors:** Keshava Praveena Neriya Hegade, Rama B. Bhat, Muthukumaran Packirisamy

**Affiliations:** Optical Bio-Microsystems Laboratory, Department of Mechanical and Industrial Engineering, Concordia University, Montreal, QC H3G 1M8, Canada; keshavapraveena.neriyahegade@mail.concordia.ca (K.P.N.H.);

**Keywords:** extracellular vesicles, exosomes, liquid biopsy, cancer prognosis, cancer diagnosis, minimally invasive diagnosis, microfluidics, nano-plasmonic detection

## Abstract

Cancer diagnosis requires alternative techniques that allow for early, non-invasive, or minimally invasive identification. Traditional methods, like tissue biopsies, are highly invasive and can be traumatic for patients. Liquid biopsy, a less invasive option, detects cancer biomarkers in body fluids such as blood and urine. However, early-stage cancer often presents low biomarker levels, making sensitivity a challenge for integrating liquid biopsy into early diagnosis. Recent studies revealed that extracellular vesicles (EVs) secreted by cells are apt markers for liquid biopsy. Detecting extracellular vesicles (EVs) for liquid biopsy faces challenges like low sensitivity, EV subtype heterogeneity, and difficulty isolating pure populations. Label-free methods, such as plasmonic biosensors and Raman spectroscopy, offer potential solutions by enabling direct analysis without markers, improving accuracy, and reducing complexity. This review paper discusses current challenges in EV-based liquid biopsy for cancer diagnosis and prognosis. It addresses the effective use of microfluidics and nano-plasmonic approaches to address these challenges. Enhancing label-free EV detection in liquid biopsy could revolutionize early cancer diagnosis by offering non-invasive, cost-effective, and rapid testing. This could improve patient outcomes through personalized treatment and ease the burden on healthcare systems.

## 1. Introduction

Cancer diagnosis typically involves a combination of radiographic imaging, surgical procedures, and chemotherapy treatments. Radiography procedures such as Magnetic Resonance Imaging (MRI), Computed Tomography (CT scan), and Endoscopic Ultrasound (EUS) are limited by the size of the lesion in the early-stage cancer. Tissue biopsy has been a reliable diagnostic technique for detecting cancer. In this method, a part of the “suspected tumor” (lesion) from the patient is surgically removed and studied for the presence of an oncogene. Hence, tissue biopsy can potentially be highly invasive based on the location of the lesion. Surgery presents a traumatic experience for the patient at the pre-diagnostic stage. Tissue biopsies capture a single sample of tumor tissue, thus ignoring the tumor heterogeneity, making this technique inadequate for frequent repetition. Furthermore, tissue biopsy cannot reflect the response to the cancer treatment because it needs time to deliver critical information [1,2]. Cancers such as Lung, Glioblastoma (Brain tumor), Pancreatic, Colon, and Rectum cancer are highly invasive. Cancer detection at an early stage dramatically improves the survival rate of patients [3].

Over the past decade, a new diagnostic method for cancer known as ‘liquid biopsy’ has been widely researched. Liquid biopsy is similar in principle to tissue biopsy but is distinct and reflects a more comprehensive tumor genetic profile than tissue biopsy. It is a clinical diagnostic procedure in which circulating cancer biomarkers in a patient’s body fluid, such as blood, urine, saliva, etc., are tested [4]. These circulating biomarkers include circulating tumor cells (CTCs), circulating nucleic acid (cNAs), Tumor-educated platelets (TEPs), and extracellular vesicles (EVs). Figure 1a shows the various types of liquid biopsy analytes present in the blood. There is an exponential increase in the number of extracellular vesicles circulating in the blood at early-stage cancer compared to other circulating biomarkers (Figure 1b). A real-time, minimal/noninvasive, and reliable tumor-specific technique that can monitor cancer’s growth and deterioration and show the response to the therapeutic effect. Liquid biopsy can enhance the reliability of clinical assessment and decision-making in cancer prognosis/diagnosis due to its inherent advantages.

We present our review in five sections. The first section introduces the multidisciplinary topic of liquid biopsy, covering currently approved tests in North America, the different classes of analytes found in blood, and the role of exosomes in cancer progression. In the second section, we briefly discuss extracellular vesicles (EVs), their subpopulations, and various cancer biomarkers associated with these vesicles. The third section focuses on the existing techniques for detecting and isolating EVs using microfluidic platforms. The fourth section explores current label-free nano-plasmonic approaches for analyzing EVs. Finally, we conclude our review in the fifth section by addressing the challenges of adapting EVs for liquid biopsy and outlining future research possibilities.

In North America, liquid biopsy tests successfully obtained clearance from the regulatory bodies of respective countries (Health Canada in Canada and the FDA in the USA). In Canada, the CELLSEARCH CTC kit (Menarini Silicon Biosystems, Inc., Huntingdon Valley, USA) has been developed as a CTC-based cancer detection kit for breast, prostate, or colorectal cancer [6]. CtDNA-based liquid biopsy tests have made noteworthy progress in cancer detection in the USA. Several “At-home test kits” have obtained approval from the FDA [7].

Liquid biopsy tests are typically performed at three distinct stages: the first stage, which often stands for prognosis or screening; the second diagnosis stage, where cancer is already confirmed with other diagnostic techniques and is in an advanced stage; and the third, which is to detect recurrence of cancer or to study responses to cancer treatment. Table 1 summarizes the state-of-the-art liquid biopsy test in North America (data may not be complete).

Although encouraging, existing liquid biopsy methods are not yet suitable for use as an independent diagnostic tool for cancer. They require validation through invasive techniques such as tissue biopsies or imaging tests. The current FDA-approved liquid biopsy assays are intended for advanced cancer stages, where patients have already received a cancer diagnosis through conventional methods. This dependence on additional diagnostic approaches highlights the need for enhanced sensitivity and specificity before liquid biopsy can become a dependable, stand-alone tool for cancer detection.

Identifying cancer-related biomarkers in blood or other bodily fluids usually demands high sensitivity, given that the levels of these biomarkers tend to be low during the early stages of cancer. Consequently, existing liquid biopsy tests often miss early-stage cancers or fail to detect them in patients with a low tumor burden, resulting in false negatives. The following section discusses the classes of analytes in liquid biopsy.

### 1.1. Classes of Analyte in Liquid Biopsy

The four classes of analytes in liquid biopsy are circulating tumor cells (CTCs), circulating nucleic acids (CNAs), circulating proteins, and circulating vesicles. Figure 1b shows the circulating levels of various tumor biomarkers in blood across various stages of cancer.

#### 1.1.1. Circulating Tumor Cells (CTC)

Circulating tumor cells (CTCs) are initially released from primary tumors in the tissue. These cells travel through the circulatory system and contribute to the development of metastatic tumors at distant sites in the body. CTCs can recirculate into the bloodstream from metastatic sites, potentially contributing to tumor spread. In terms of their prevalence, CTCs are relatively rare, with approximately one CTC found for every million leukocytes (white blood cells) in the blood.

CTCs have become highly significant for tumor detection and are increasingly used as a non-invasive alternative to traditional tissue biopsies. They are easier to sample and can provide real-time data about tumor conditions. Research has shown that CTC levels can change more dynamically and are often more reflective of tumor status than conventional blood biomarkers [20].

#### 1.1.2. Circulating Cell-Free Nucleic Acids (Cf-NAs)

Cell-free nucleic acids (cfNAs) originate from cells through cell death. They include circulating cell-free DNA (cfDNA) and circulating cell-free RNA (cfRNA). cfDNA released from tumor cells is called circulating tumor DNA (ctDNA). They are found in body fluids such as blood. Low availability in early-stage cancer, the possibility of chemical damage to the molecules, and lower half-life in the blood make ctDNA less suitable for early-stage liquid biopsy [21,22,23]. However, they can be used in the post-operative stage [24].

#### 1.1.3. Tumor-Educated Platelets (TEPs)

Platelets are tiny fragments of cells. Platelets internalize tumor RNAs, which alter the platelets’ functioning and enhance tumor metastasis through the blood [25,26,27].

#### 1.1.4. Exosomes

Exosomes are nano-sized vesicles secreted by cells. They belong to a much broader vesicle family known as extracellular vesicles (EVs). EVs comprise other types of vesicles, such as Microvesicles (MV), Ectomeres, Apoptotic bodies (AB), etc. [28]. The exosome is a critical player in cancer metastasis. Their role in disease progression has been studied extensively [29,30]. The nanosized vesicles transport cell cargo such as miRNA, mRNA, proteins, etc. When a cell undergoes some abnormal activities, these changes are translated via exosomes and other extracellular vesicles. For instance, the secretion of exosomes increases multi-fold when a healthy cell is under the invasion of cancer [31,32]. Also, exosome content varies as the cell undergoes foreign invasion [33]. As shown in Figure 1b, exosomes are available at higher concentrations at the initial stages of cancer. Hence, exosomes are the most suitable analyte for an early-stage cancer diagnosis through liquid biopsy.

### 1.2. Hurdles in Adapting Exosomes in Liquid Biopsy

Exosomes form a significant subpopulation of EVs. However, isolating exosomes from the mix of other vesicles is challenging. Studying exosomes at the molecular level is strenuous for three reasons. (a) Isolation of exosomes is tedious. The ultracentrifugation method is a commonly used method for the isolation of EVs. This method involves a sequence of centrifugations at progressively higher relative centrifugal forces (RCFs) (from 10^3^× *g* to up to 10^5^× *g*). As such, ultracentrifugation yields are low due to successive centrifugation. Furthermore, a large sample volume is required. (b) Recovery of isolated exosomes for analysis without losing their content. (c) Sensitive detection of cancer biomarkers contained in the exosomes.

### 1.3. Handling and Analysis of Exosomes: A Microfluidic Approach

Microfluidic techniques are excellent platforms to navigate the obstacles above. They offer precise manipulation of many bioparticles, small-volume capacity, efficient mass and energy transport, and high levels of process integration. Passive and active microfluidic methods have been utilized for label-free, high-resolution separation of EVs from the biological matrix. Precise manipulation of small volumes and selective separation of extracellular vesicles (EVs) ensures the functional molecules of EVs are intact. Microfluidic platforms revolutionize biosensors. Microfluidic isolation techniques function under controlled microenvironments, which helps to maintain the integrity of the biological properties and functionality of the isolated subtypes [34]. Microfluidic channels can be designed with various geometries to obtain specific flow rates. This ensures minimal shear stress and provides high specificity in capturing target subtypes. Hassanpour Tamrin, S. et al. [35] have reviewed label-free microfluidic technologies for exosome isolation. These label-free microfluidic protocols can be performed rapidly and straightforwardly without hampering EVs’ biological activity and functionality. Microfluidic platform-based detection strategies have shown promise in becoming exemplary sensors with high efficiency and specificity and a low limit of detection (1 ng/mL) [36]. Furthermore, microfluidic platforms seamlessly integrate optical and non-optical detection and analysis techniques.

## 2. Extracellular Vesicles: Exosomes, Microvesicles, and Others

Extracellular vesicles (EVs) are small membrane-bound particles secreted by cells into the extracellular space. They play key roles in intercellular communication and can carry a variety of bioactive molecules, including proteins, lipids, nucleic acids (like RNA and DNA), and metabolites [37,38,39]. Exosomes (50–200 nm), Microvesicles (50–1000 nm), and Apoptotic bodies (1–5 µm) comprise the majority of EVs [40]. Figure 2 depicts the biogenesis of EVs.

### 2.1. Exosomes

The biogenesis of exosomes begins with endosomes. Endosomes are organelles formed due to endocytosis, a process where the cell engulfs extracellular material by invaginating the plasma membrane, as shown in Figure 2. Newly formed endosomes are called early endosomes, while the mature ones are called late endosomes [41]. Early endosomes are formed immediately after cargo internalization via endocytosis and act as sorting compartments. Membrane proteins, lipids, and other cellular cargos are directed to different cellular pathways. As early endosomes mature, they become late endosomes (or multivesicular bodies, MVBs), which contain intraluminal vesicles (ILVs). These ILVs are crucial for the formation of exosomes. The ESCRT (Endosomal Sorting Complex Required for Transport) machinery plays an essential role in sorting cargo into these vesicles, marking the early endosomes as vital sites for the initial steps of EV biogenesis.

Late endosomes, or MVBs, further mature and carry out the final cargo sorting before it is either degraded by lysosomes or packaged into EVs for secretion. When MVBs fuse with the plasma membrane, they release their ILVs as exosomes into the extracellular space. These exosomes, which carry proteins, lipids, and RNAs, like microRNAs, play significant roles in cellular communication, including in processes like immune modulation and cancer metastasis. Exosomes were described for the first time in the 1980s [42]. When measured by transmission electron microscope, they have a characteristic saucer-like shape or deflated sphere [43]. Exosomes are released via two pathways. The first pathway involves the formation of Intraluminal Vesicles (ILVs) within Multivesicular Endosomes (MVEs). In turn, the membrane of MVE fuses with either the lysosome for cargo degradation or the plasma membrane, resulting in the release of ILVs. Once secreted, ILVs are called exosomes. The second pathway is called the “direct pathway”; T cells and erythroleukemia cell lines release exosomes from the plasma membrane spontaneously after surface receptors cross-linking.

Exosomes are involved in the cell’s biological activities, such as intercellular communication, cargo delivery, etc. [44,45]. Inhibition of RAB27A, a protein known for trafficking exosomes, reduces the trafficking of exosomes in MCF7 and MDB MA 453 cells [39]. Exosomes are nano-vehicles at the cellular level. Exosomes secreted by cancerous cells are known to be involved in disease progression. They enter healthy cells and inhibit the release of healthy exosomes from healthy cells [46]. Exosomes interact with other cells via exchanging biomolecules, including messenger RNA (mRNA), microRNA (miRNA), DNA, proteins, etc. Some of the common biomolecules found in exosomes are shown in Figure 3.

Exosomes are widely studied EVs. Some pieces of literature use the terms EV and exosome interchangeably, while others refer to exosomes as small extracellular vesicles (sEVs).

Exosomes play a crucial role in reshaping the tumor microenvironment (TME). They carry bioactive molecules such as proteins, lipids, RNA (miRNAs, mRNAs, and lncRNAs), and metabolites. This cargo influences the behavior of recipient cells. Exosomes secreted by cancer cells enhance tumor growth and metastasis by promoting angiogenesis, stimulating tumor cell migration and invasion, and inducing epithelial-to-mesenchymal transition (EMT) [47,48]. For example, exosomal miRNAs, like miR-21 and miR-23, suppress tumor suppressor genes and increase tumor aggressiveness [49]. Exosomes mediate immune suppression by transferring immune-modulatory molecules such as PD-L1, TGF-β, and cytokines to immune cells. This promotes immune tolerance and prevents effective anti-tumor immunity. Exosomes from primary tumors prepare distant organs (such as the lungs, liver, or brain) to become more susceptible to metastatic colonization. They transfer pro-inflammatory factors, matrix metalloproteinases (MMPs), or other molecules that facilitate tumor cell adhesion and invasion in the target organs.

### 2.2. Microvesicles

Microvesicles were first described by Wolf in 1967 [50]. They are released from the plasma membrane during periods of cell stress through a process known as budding, followed by the fission of the plasma membrane into the extracellular space. This release is driven by dynamic interactions between the redistribution of phospholipids and the contraction of cytoskeletal proteins in response to various stimuli such as apoptosis, hypoxia, cellular activation, etc. These microvesicles, often called microparticles, are present in most, if not all, biological fluids and conditioned culture media. The typical diameter of these MVs is between 500 and 1000 nm, and they are heterogeneous. Their density is in the range 1.16–1.26 g/mL. The size ranges of microvesicles and exosomes may overlap, primarily when body fluids are used to isolate vesicles. Although microvesicles and exosomes are distinct types of EVs, neither size, morphology, nor exposure to phosphatidylserine (PS) is a sufficient criterion to distinguish two kinds of EVs from each other. Microvesicles are secreted by outward budding, as shown in Figure 2. Table 2 lists the properties of different EV types. On the one hand, exomers are non-EV structures present in EVs after isolation due to their common physical properties. Lipoproteins are protein biomolecules with different densities. They are named after their densities (VLDL-very low-density lipoprotein; LDL—low-density lipoprotein; IDL—intermediate-density lipoprotein; HDL—high-density lipoprotein) [51,52].

MVs serve as vehicles for direct intercellular communication by transferring surface proteins, such as miRNAs, and other cargo that modulate the phenotype of recipient cells. They play a role in exchanging oncogenes and tumor suppressor genes between tumor cells and surrounding cells. MVs contribute to cancer progression by directly transferring oncogenic protein p53 [53] or signaling molecules like growth factors (e.g., VEGF, EGF) to neighboring cells, thereby promoting tumor cell survival, proliferation, and migration. MVs released by tumor cells impair the function of endothelial cells and assist in forming new blood vessels through angiogenesis, which is crucial for tumor growth and metastasis. MVs also promote blood clotting by carrying tissue factor (TF) and other procoagulant proteins. This enhances tumor-induced thromboembolism, which is associated with poor prognosis in cancer patients. MVs promote thrombus formation and immune evasion, creating a conducive environment for metastasis.

**Table 2 ijms-26-06352-t002:** Classifications of diverse types of EVs. (Data compiled from [54]).

	Exosomes	Microvesicles	Apoptotic Bodies
Size	20–100 nm	50–1000 nm	500–2000 nm
Density	1.13–1.19 g/mL	1.04–1.07 g/mL	1.16–1.28 g/mL
Biogenesis	Formed through inward luminal budding of the membrane and fusion of multivesicular bodies with the cell membrane (endolysosomal pathway).	Directly shed from the cell’s plasma membrane through outward budding.	Formed through membrane blebbing from a cell undergoing apoptosis.
Composition	The membranes have an elevated level of amino phospholipids and lipid ceramide compared to the outer leaflet of the plasma membrane. Lipid ceramide plays a key role in the membranes. The production of ceramide is an essential step in the sorting and generation of exosomes.	Lipid composition is similar to the cell membrane but lacks the asymmetric distribution of lipids. Amino phospholipids, phosphatidylserine, and phosphatidylethanolamine are not sequestered. A higher concentration of cholesterol and sphingolipids compared to the cell membrane.	Externalization of phosphatidylserine is distributed on the cell surface. Presence of Annexin I and calreticulin. May also include FasL/FasR, TNF-α/TNFR1, Apo3L/DR3, Apo2L/DR4, and Apo2L/DR5.
Biomarker	Tetraspanins family (such as TSPAN29 and TSPAN30, CD81, CD82, CD9, CD63), ESCRT proteins (Alix, TSG101), actin, flotillin, Hsc70, HSP 90, Hsp60 and Hsp20 clathrin, integrins (such as α3, α4, β1, β2)	Integrins, selectins, flotillin-2, CD40 ligand, metalloproteinase	Annexin V positivity, phosphatidyl serine

### 2.3. Apoptotic Bodies

When the cells undergo apoptosis, apoptotic bodies (AB) or vesicles are released [55]. They contain cell organelles and nuclear fractions. ABs are significantly larger than exosomes, with a diameter of 1–5 μm, indicating that their size falls within the range of platelets in human blood. Apoptotic bodies play a role in both immune activation and suppression. For example, apoptotic bodies activate dendritic cells, stimulating the anti-tumor immune response. However, when released by tumor cells, they also transfer immunosuppressive factors that aid in immune evasion. Apoptotic bodies influence the initiation of immune responses by presenting tumor antigens to immune cells, influencing the initiation of immune responses. The contents of apoptotic bodies, such as cellular debris, miRNAs, etc., influence the behavior of surrounding cells in a tumor microenvironment [54]. While they act as a means for tumor cells to disseminate information, their precise role in promoting metastasis is still under investigation.

Like other members of the EV family, the density of ABs partly overlaps with that of exosomes, with values ranging between 1.16 and 1.28 g/mL. Table 2 lists the biophysical characteristics of major EV subtypes, namely, exosomes, MVs, and ABs. A comprehensive list of biomarkers associated with EVs is given in Table 3.

## 3. Microfluidic Platform for EV Isolation, Detection, and Characterization

### 3.1. EV Isolation

Ultracentrifugation is widely regarded as the ‘gold standard method’ for isolating EVs. Before ultracentrifugation, biofluids containing EVs need to be enriched [103]. Commonly used methods for EV enrichment before ultracentrifugation include low-speed centrifugation and filtration. Due to the multiple enrichment stages involved, the yield of recovery in the ultracentrifuge technique is typically less than 25%. This method requires a large sample volume. Ultracentrifuge equipment is expensive, and consumables such as tubes and reagents add to the overall cost of operations. The process is time-consuming, taking up to 16 h.

Other EV isolation techniques include size exclusion chromatography (SEC), filtration, and immunoaffinity-based techniques. SEC and filtration co-isolate smaller particles, while immunoaffinity-based techniques are less suitable for downstream processing. However, these techniques are affordable in a clinical setting.

In contrast, numerous commercial EV isolation kits significantly reduce EV isolation time to less than 4 h. However, these kits pose the risk of co-isolating other EV-like particles, and the isolated EVs may be less compatible with downstream processing [103].

On the other hand, the microfluidic platform provides a more straightforward method and can be used conveniently with much smaller sample volumes (in the order of microliters). Microfluidic techniques can utilize physical properties, such as size, which may also result in the co-isolation of EV-like particles, similar to ultracentrifugation. Additionally, isolating EVs based on biological properties, such as surface markers, allows for subtype-specific isolation of EVs. In clinical settings, microfluidic techniques have an advantage over ultracentrifugation. Although the initial cost of microfluidic devices can be high, they offer the benefit of scalability and automation-friendly processes that can lower operational costs in the long run. Adapting microfluidic platforms in EV analysis offers a comprehensive solution (“one pot”) for label-free EV detection, isolation, and characterization in a single setup. The following section will explore EV isolation using microfluidic platforms.

#### 3.1.1. Physical Technique for EV Isolation in a Microfluidic Platform

Size-based separation of EVs from serum samples using elastic lift yields a higher population of smaller EVs (<200 nm). Heterogeneous EVs confined to a straight microfluidic channel are subjected to drag and elastic force by introducing a Newtonian fluid, which contains heterogeneous EVs on the side and a sheath fluid in the middle, as shown in Figure 4. Due to the force imbalance, larger EVs will be drawn to the middle stream, thus creating two distinct streams of EVs with assorted sizes. The purity and recovery of the method were found to be >90% [104]. This method is further extended to obtain different size-based fractions of EVs by employing linear λ-DNA and aptamer [105].

A viscoelastic-based microfluidic platform has been developed for the label-free isolation of extracellular vesicles (EVs) from whole blood. This is accomplished through the use of inertial, viscous, and drag forces. In this process, a sample consisting of whole blood and phosphate-buffered saline (PBS) as a diluent is passed through a specially designed microfluidic channel. Smaller EVs are directed along the walls of the microfluidic channel, while larger particles flow along the center. This method achieves a recovery rate of 87% and a purity of 97% [106]. Pulsating membrane filtration was achieved in a microfluidic channel for isolating extracellular vesicles from blood samples of clinical patients. EV yields between 76 and 92% were achieved by this method in 30 min [107].

#### 3.1.2. Asymmetric Flow Field-Flow Fractionation (A4F)

Asymmetric Flow Field-Flow Fractionation (AF4 or A4F) is a label-free separation technique that can be used to isolate particles based on their hydrodynamic size within a fluidic channel [108]. The process begins by injecting the sample into a flow channel where a non-uniform cross-flow is applied. This asymmetry in flow creates a gradient of resistance across the channel, which forces particles toward the center where the flow is least affected, enabling hydrodynamic focusing. In this focused flow, particles are separated based on their size: smaller EVs (like exosomes) accumulate near the walls of the channel, while larger EVs (like microvesicles) move toward the center. As the sample progresses through the channel, size-based separation occurs due to the different resistance forces acting on varying-sized particles. The different EV populations elute from the channel at different times, and each population is collected in separate fractions.

AF4 provides high-resolution separation without the need for labels or reagents, thus making it a gentle method that preserves the integrity of the EVs. Moreover, the technique is versatile, separating a wide range of EV sizes, making it suitable for isolating exosomes, microvesicles, and apoptotic bodies from biological fluids like blood, urine, or cell culture media.

#### 3.1.3. Electrical Technique for EV Isolation in Microfluidic Platform

Electrical techniques for EV isolation involve techniques using the electrical properties of EVs, such as charge and the dielectric constant. An external electric field can be used in this technique. One of the simplest electrical techniques for EV isolation is a charge-based separation. This technique relies on the surface charge of the EVs.

EVs, in general, possess a negative surface charge due to the chemical composition of their lipid bilayer, which includes phospholipids, cholesterol, and surface-associated proteins. The hydrophilic headgroups of the phospholipids carry negative charges, which impart a net negative surface charge to most EVs. However, the exact charge can vary depending on the cell type of origin, the physiological environment, and the presence of specific proteins or glycoproteins on the surface. Similarly, EVs have a negative Zeta potential, a non-firmly held charge covering the stern layer resulting from the particle’s surface charge. Typically, EVs’ zeta potentials range between −10 and −50 mV, depending on their origin and purification method [109].

Thakur, A. et al. [110] have shown that EVs are electrostatically adsorbed onto a positively charged self-assembled monolayer (SAM) gold nano-island. Exosomes are firmly held to the AuNI-SAM. Exosomes are known to have a higher negative zeta potential compared to microvesicles.

Another electrical technique for EV isolation is using the dielectric constant of the EVs. The dielectric constant governs the rate of charge movement within the material for an external electrical field. An alternating current electrokinetic microarray (ACE) chip was fabricated by Ibsen et al. [111]. The chip provides a high and low dielectrophoretic (SEP) field with an application of an alternating current, as shown in Figure 4B. In this chip, separation is achieved due to size variations in the EVs. Larger particles are pulled towards the DEP low field, while smaller particles are unaffected. This technique achieved a recovery rate of >50%. Xing, Y. et al. developed circular multi-cavity electrophoresis for EV isolation with a recovery rate of around 87% [106]. The variation in mobility among EV subpopulations under an electric field was used to separate EVs through two-dimensional electrophoresis. Micro-slit-well structures arranged periodically enabled the fractionation of EV subtypes [112].

Label-free, contact-free EV isolation in a microfluidic platform was performed using the Optically induced Dielectrophoresis (ODEP) technique. EVs are manipulated using DEP forces. A recovery rate of 52.2 ± 8.6% was achieved [113].

Physical characteristics like size and density overlap in most extracellular vesicle (EV) subpopulations. As a result, when employing isolation methods that rely on the physical attributes of extracellular vesicles (EVs), there is a possibility of co-isolating distinct EV subpopulations, thus reducing the efficacy of the isolation process. The primary EV subpopulations, microvesicles, and exosomes are typically distinguished based on their surface markers, highlighting the importance of immunoaffinity-based techniques for precise EV isolation.

#### 3.1.4. Immunoaffinity-Based Techniques for EV Isolation in Microfluidic Platform

Gwak, H. et al. [114] developed a modular microfluidic platform designed to isolate extracellular vesicles (EVs) from cell culture samples. The microfluidic device consists of two main sections: a mixer and a trapper. In the mixer, EVs are selectively captured using anti-CD63 functionalized microbeads. The second section, the trapper, employs a fish-trap-shaped microfilter to retain the EVs bound to the microbeads. This rapid method allows for EV isolation, with the entire process lasting just 5 min. The device achieved a high efficiency rate of 97.18%.

Extracellular vesicles (EVs) from human plasma were isolated using a magnetic levitation method. First, the EVs were bound to three types of streptavidin-coated polymer beads, each functionalized with different antibodies: CD9, CD63, and CD81. The functionalized polymer beads were then incubated with the EVs overnight to capture EVs. Following this, the mixture was passed through a microfluidic channel while under a magnetic field. The nonmagnetic polymer beads respond to a magnetic field in a density-dependent manner. The density-based isolation of the EVs was achieved through this magnetic levitation process [115].

### 3.2. EV Detection

Due to their size range, EV detection techniques require high resolution and lower detection limits. Current EV detection techniques are broadly classified as optical techniques and non-optical techniques, and are listed in Table 4 [116].

Optical methods of EV detection utilize the optical phenomena that occur when light hits the object, such as scattering, absorption, transmission, etc. They offer LOD up to atto mole (10^−18^ M). Optical methods are further classified as scattering, fluorescence, and plasmonic-based. The first optical method, scattering techniques, depends on the ability of the sample to scatter light, the sample’s size, and the sample’s concentration. Optical detection setups are equipped to detect scattered light. The amount of light scattered depends on a factor called “scattering cross section”, which is given by Equation (1) [116].(1)$$\sigma \propto \frac{d^{6}}{\lambda^{4}}{\left(\frac{m^{2}-1}{m^{2}+2}\right)}^{2}$$
where *σ* is scattering cross section, *d* is the particle diameter, *λ* is wavelength of incident light and m is the refractive index ratio of sample to medium. At 532 nm, a typical optical device wavelength, Rayleigh’s approximation is about 50 nm. Hence, smaller samples are difficult to detect with scattering techniques.

The second optical method, fluorescence, involves the property of a material that absorbs light of a particular wavelength and re-emits it at a longer wavelength. EVs do not inherently possess fluorescence and must be tagged with fluorescent tags. Thus, fluorescent techniques are not label-free.

The third optical method uses plasmons to detect samples. When a light (photon) hits a metal at a certain angle, the surface plasmon resonance angle (θ_SPR_), free electrons at the metal–dielectric surface gain energy from the incident photon, enter an excited state, and oscillate. These excited free electrons at the metal surfaces are called surface plasmons, and the collective oscillation of plasmons is called Surface Plasmon Resonance (SPR), which creates an evanescent field consisting of electromagnetic waves propagating along the metal surface. Typically, these evanescent fields have a depth of a few hundred nm from the metal–dielectric surface. Their intensity depends on the dielectric medium. Evanescent fields are sensitive to any changes to the dielectric medium. When a biomolecule attaches to the metal surface, it causes a change in the dielectric medium, thereby changing the local refractive index and the evanescent field. The sensitivity of the plasmonic biosensors is expressed in terms of the refractive index unit (RIU). Sensors with smaller RIU can detect very minute changes in refractive index. Typical plasmonic sensors have RIUs from 10^−3^ to 10^−6^. Thus, plasmonic techniques offer label-free detection and analysis.

Plasmonic techniques require fewer reagents than scattering or fluorescent labeling techniques [117,118]. They are commonly used for studying kinetics and affinity-related studies. They include Surface-Enhanced Raman Spectroscopy (SERS), Surface Plasmon Resonance (SPR), and Localized surface plasmon resonance (LSPR) [119]. The following section discusses these techniques that are suitably adapted to EV analysis.

## 4. Plasmonic Technologies for Exosome Analysis

### 4.1. Raman Scattering

Raman spectroscopy is an optical technique used to analyze the molecular composition (both in a static and dynamic state) of biological samples, known as the analyte. In Raman scattering, photons from a light source scatter inelastically from molecules in the analyte, either gaining or losing energy equivalent to the vibrational levels of the molecules Figure 5a. Consequently, the scattered photons have a different wavelength from the incident photons. The difference in energy between the incident and scattered photons is caused by the molecular bonds present in the analyte. Each type of molecular bond possesses characteristic vibrational frequencies or energies. Detecting all scattered photons generates a Raman spectrum plotted on the Cartesian coordinate. Abscissa indicates the wavenumber or Raman shift (cm^−1^), and Ordinate indicates the Raman scattering intensity. The Raman spectrum provides an elaborate and multiplexed chemical composition of biological samples [120]. The Raman signal is dependent on the Raman cross-section. It is an effective area where photons must interact with molecules to undergo Raman scattering.

The intensity peaks in the Raman spectrum are directly related to molecular concentration. With its high spatial resolution, Raman spectroscopy is a non-invasive and label-free method, making it an excellent technique for molecular characterization of EVs. However, the signal strength of Raman spectroscopy is contingent on the number of scattered photons from the sample. Typically, only one in a million incident photons will scatter, producing relatively weak Raman scattering signals [121].

Raman spectroscopy is not well suited for analyzing biological samples in suspension, such as EVs. A Raman signal is impaired by the background signal originating from the liquid present in the suspension. Raman spectroscopy combined with optical trapping is suited for analyzing samples in suspension. This technique is called Laser Tweezer Raman Spectroscopy (LTRS). In this method, an analyte is trapped and immobilized within a focal volume using a laser trap created by a tightly focused Gaussian laser beam. This selective trapping enhances the signal-to-noise ratio by focusing on the target analytes while excluding non-relevant signals. As a result, it allows for more accurate detection and analysis of specific biomolecules or particles in complex samples. Optical trapping and excitation of Raman signals from the trapped analyte co-occur from a single laser source, and the events are detected using a spectrometer and a confocal detection setup. The laser focal volume for generating Raman signals is approximately 1 μm^3^. Due to its smaller focal volume, LTRS is well-suited for analyzing EVs. Moreover, LTRS is rapid and non-invasive and has been utilized to differentiate EVs from diverse sources based on membrane content [122,123]. The results obtained from LTRS are the average value of all the analytes analyzed. Hence, the homogeneity of EVs must be ensured before using LTRS for analysis.

### 4.2. Surface-Enhanced Raman Scattering

Surface Enhanced Raman Spectroscopy (SERS) is a spectroscopic method in which the Raman signal is enhanced using a metal in a substrate or nanotags (for the analyte in a liquid sample), as shown in Figure 5b. SERS enhances Raman signals through two simultaneous phenomena: electromagnetic excitation and chemical excitation, with the former being in the order of 10^10^ and the latter in the order of 10^2^. The enhancement of the Raman signal through electromagnetic excitation is due to surface plasmons formed when the laser light hits the metal surface. Thus, the surface plasmons formed offer a stronger magnetic field locally to the molecule attached to the metal and increase the Raman cross-section, amplifying the Raman scatter signal. SERS is a sensitive method capable of enhancing Raman signals by orders of 10^15^. It is effectively employed for protein profiling of exosomes [124]. Surface modifications of the substrate, which can be physical and chemical, have been shown to increase the Raman signals. The following section discusses some of the physical and chemical modifications of the surface in SERS-based EV analysis.

#### 4.2.1. Physical Modification of SERS Substrate

Physical modification of Surface-Enhanced Raman Spectroscopy (SERS) substrates involves altering the substrate’s surface morphology to enhance Raman signals. This can include processes such as nanoparticle deposition, roughening, or creating nanostructures like nanorods or nanospheres. These surface modifications increase the density of localized surface plasmon resonances, which enhances the sensitivity and specificity of SERS. Physical changes like these allow for better detection limits and improved signal intensity.

A superhydrophobic surface (SHS) is created by fabricating silicon micropillars, as shown in Figure 6. Reactive ion etching of a silicon substrate covered with circular silver masks produces micropillars. A silver mask to produce silicon micropillars is created by the electrodeless deposition of silver onto a patterned photoresistor on a silicon substrate. The standard lithographic method created a patterned photoresistor on a silicon substrate. SHS repels water; naturally, the analyte is localized on the substrate. Raman peaks of exosomes derived from healthy cells contained more significant amounts of lipid, with the predominant peaks at 850 cm^−1^ and 1038 cm^−1^, and exosomes derived from cancerous cells contained high RNA content with Raman peaks at 1235 cm^−1^ and 1278 cm^−1^ [125].

A SERS substrate featuring a graphene layer-enclosed gold pyramid structure was utilized to assess the molecular composition of exosomes derived from four distinct sources [126]. The sources of extracellular vesicles (EVs) examined included those from fetal bovine serum, human serum, and the cell culture media of two lung cancer cell lines (HCC827 and H1975). The gold pyramids were fabricated using a nano-casting technique, wherein gold layers were deposited onto a silicon surface with pits created through plasma etching, followed by the anisotropic etching of polystyrene-patterned silicon wafers. These structures generate an intense surface plasmonic field. The graphene layer offers a chemically stable and biocompatible surface for sensing applications. In this study, EVs derived from serum, specifically fetal bovine serum (FBS) and human serum, showed a higher relative intensity for nucleic acid bands compared to EVs obtained from cancer cell lines. In contrast, EVs from cancer cell lines exhibited a greater relative intensity for lipids.

A glass surface coated with polystyrene particles produces patterned PDMS subjected to Silver (Ag) sputtering to obtain nano bowl structures. Exosomes from ovarian cancer (SKOV3) conditioned cell culture media were isolated using ultracentrifugation and a total exosome isolation reagent kit (TEIR). Isolated exosomes were dropped onto a substrate containing silver nano bowls and dried before taking SERS data. The delicate membrane of EVs that holds the internal biomolecules bursts when exposed to a strong light source. SERS data of exosomes obtained from two different techniques show some discrepancy in peak position and intensity, pointing to the bias of each technique towards exosomes in conditioned media [127].

#### 4.2.2. Chemical Modification of SERS Substrate

Chemical modification of the metallic substrate enhances signal as well as specificity. LXY30 peptide is used to combine silver and exosomes. α3ß1 integrin is a member protein overexpressed in human ovarian cancerous cell SKOV-3 and its released exosomes. The specificity of LXY30 to interact with α3ß1 integrin makes this a valuable method in detecting cancerous exosomes derived from SKOV-3 cells. Ag nanoparticles were functionalized with LXY30 by incubation for 2 h. LXY30 functionalized Ag nanoparticles were incubated with exosomes for 18 h, and the resulting solution was centrifuged. The sample was tested on a glass slide substrate by dropping a single drop and allowing it to dry [128].

Selective capturing of EVs from liquid samples is conducted by electrostatic adsorption. The cationic amino group present in cysteamine imparts a positive charge on gold aggregates on a glass substrate functionalized with cysteamine. Exosomes carry negative charges on the surface. The anionic surface of an exosome from the liquid sample is detected and captured by a positively charged substrate [129]. Further studies on the SERS-based detection of cancerous EVs are reviewed by Guerrini et al. [130].

Rojalin, T. et al. have [131] developed a SERS-based platform for liquid biopsy by utilizing a borosilicate scaffold with silver nanoparticles (AgNP). The borosilicate scaffold serves as a filter to trap smaller EVs (50–200 nm), and the silver particles enhance the Raman signal by an order of 10^5^, offering a novel method for the isolation and detection of EVs. As a result, the chemical composition of ovarian cancer and endometrial cancer has been successfully obtained. Notably, EVs from ovarian malignancy (Ov-Ca I–III) and trypsinized EVs from serous endometrial cancer patients exhibited distinct bands at 904, 1287, 1336, and 1450 cm^−1^. The limit of detection of the device was 600 particles/mL when tested with SKOV-3 EVs.

Electrostatic adsorption of cationic gold nanoparticles in colloidal solution to anionic EVs was explored by Stremersch, S. et al. [124]. Exosome-like vesicle ELVs, called EVs elsewhere, were isolated from red blood cells (RBC) and B16F10 melanoma cancer cells. They were coated with colloidal gold via electrostatic adsorption. It was estimated that approximately 600 AuNPs were required to ‘pack’ individual ELVs derived from B16F10 melanoma cancer cells and 1200 AuNPs for ELVs derived from RBCs. The higher number of AuNPs needed is primarily due to the higher surface area of exosomes from RBCs. The adsorbed AuNPs enhance the Raman signal by generating localized surface plasmons. The biomolecular exosome components were identified at 1123 cm^−1^ (lipids + proteins), 1172 cm^−1^ (proteins), 1307 cm^−1^ (proteins + lipids), 1366–1370 cm^−1^ (phospholipids + carbohydrates), 1445 cm^−1^ (lipids + proteins), and 1572–1576 cm^−1^ (nucleic acids).

The plasmonic nanoparticles carrying Raman reporters, known as SERS nanotags, enhance the Raman signal, provide high sensitivity, and detect the analyte. Notably, non-fluorescent SERS Nanotags enhance SERS signal intensity by 18% in the event of Raman scattering [132]. Functionalized magnetic particles enhance the Raman signal. Using MNPs can produce an enhancement factor of 225 [133].

#### 4.2.3. SERS Signal Detection Analysis

Surface-enhanced Raman scattering (SERS) signals, like Raman spectroscopy, are susceptible to background noise. The overlapping Raman signals of the analyte molecules can broaden the Raman peaks, leading to ambiguity in the results. However, our meticulous spectral data analysis, aided by the in-built data acquisition software that effectively removes most background noise, ensures the precision of our findings. Statistical tools such as Multivariate Curve Resolution with alternating least squares (MCR-ALS) or Partial Least Squares Discriminant Analysis (PLS-DA) are used to interpret the Raman spectrum, further enhancing the accuracy of the results.

Principal component analysis (PCA) of SERS data of exosomes derived from four different classes (Lung cancer (H1299 and H522) cell line, alveolar cell, and healthy cells) classifies four distinct groupings. The H1299 and H522 exosomes are classified from the rest by pattern matching the Raman peak achieved from Principal Component Analysis [134]. Principal component differential function analysis (PC-DFA) of SERS data obtained from patients diagnosed with early-stage (IA-IIB) pancreatic cancer and healthy controls can be distinguished with 90% accuracy [135]. A review of EV detection using SERS and signal analysis strategies can be found here [136].

While it is seldom seen, Jalali, M. et al. [137] developed a label-free microfluidic device integrated with SERS. A nanosurface microfluidic device based on embedded nano bowtie antennas (nano bowties) placed at the bottom of the fluidic chamber is used for SERS detection and the identification of tumor EVs. They tested EVs isolated from non-cancerous glial cells (NHAs) and two human glioma cell lines with different aggressive properties (U373 and U87) in the device. A patterned thin plasmonic film of ZnO (60 nm) and 20 nm Ag, Au, and Al layers was deposited. ZnO was used for its biocompatibility. The PCA of the SERS signal data obtained from the EV samples successfully identified EVs from non-cancerous glial cells (NHAs) from cancerous EVs obtained from U373 and U87.

Another microfluidic device integrated with SERS has been developed by Han, Z et al. [138]. This microfluidic platform efficiently isolates exosomes directly from human plasma. The system utilizes surface-enhanced Raman spectroscopy (SERS) to profile exosomal biomarkers, achieving a detection limit as low as two exosomes per microliter. Initially, extracellular vesicles (EVs) are bound to SERS tags using specific EV markers, and the samples are subsequently analyzed. The entire process can be completed within five hours using only 50 μL of plasma, providing a rapid and minimally invasive alternative to traditional biopsy methods. This integrated approach demonstrates high sensitivity and specificity for diagnosing osteosarcoma, highlighting its potential for clinical applications.

Most Raman techniques are stand-alone and less suited to integrating into microfluidics. However, the liquid biopsy workflow demands label-free techniques to analyze analytes in liquid and can be integrated into microfluidic devices. In this context, the role of SPR in liquid biopsy is crucial, providing the latest advancements in cancer research.

### 4.3. Surface Plasmon Resonance

Surface plasmon resonance (SPR) is an analytical tool for affinity binding analysis (static and dynamic), such as antibody–antigen, ligand–receptor, and enzyme–substrate reactions. Surface plasmon resonance detects local refractive changes due to the target molecule binding to the metal surface [139]. As previously mentioned, when an incident light hits a metal surface at a certain angle (θ_SPR_), electrons at the metal–dielectric surface become excited, and these excited electrons are called plasmons. Plasmons travel parallel to the metal–dielectric medium, forming an electromagnetic field called an evanescent field that decays exponentially, as shown in Figure 7. Equation (2) shows that the surface plasmon resonance angle (θ_SPR_) and local refractive index are related.(2)$$\theta_{SPR}=\sin^{-1}\left(\frac{1}{n_{1}}{\left(\frac{n_{2}^{2}n_{g}^{2}}{n_{2}^{2}+n_{g}^{2}}\right)}^{\frac{1}{2}}\right)$$
where θ_SPR_ is the surface plasmon resonance angle and n_1_ is the refractive index of the prism. n_2_ is the refractive index of the metal film (gold layer). *n_g_* is the refractive index of the cover glass. θ_SPR_ is a function of *n*_2_ when *n*_1_ and *n_g_* are fixed. During the sensing phenomena, *n*_2_ (refractive index of the gold) changes due to binding (adsorption) or non-binding (desorption).

The photodetector in the SPR instrument detects the intensity of the light reflected from the surface at the angle of incident of the light. The reflected light intensity is the least for θ_SPR_. A minor change in the refractive index at the sensing medium (due to affinity binding) results in a ‘shift’ in θ_SPR_ that could be identified from the Sensogram. The sensitivity of SPR biosensors is reported to be related to a change in the fluid medium’s refraction index (RUI). RUI is equivalent to θ_SPR_ shift of 10^−4^°. A typical value of SPR biosensors is 10^−3^ RUI, while excellent sensors have 10^−4^ RUI [141].

SPR is used in a time-dependent manner to study binding kinetics. The intensity of the reflected light is analyzed concerning time at a constant angle of incident light (θ). The binding kinetics of the EV subpopulations (Microvesicles and exosomes) are investigated using SPR with gold nano-islands (AuNIs). Nano gaps between aggregates cause a strong electromagnetic field. AuNIs can be used with or without functionalization to detect EVs. Bare AuNIs, i.e., without any functionalization, are used to distinguish exosomes and microvesicles. Exosomes directly interact with gold due to the anionic surface (high zeta potential) of exosomes [110].

Surface plasmon resonance was used to analyze exosomes derived from the serum of multiple myeloma (MM) patients, monoclonal gammopathy of undetermined significance (MGUS) patients, and healthy individuals. MM cells are characterized by the proliferation of a single clone of plasma cells derived from B cells in the bone marrow. MM-derived exosomes increase the survival and expansion of tumor cells by shuttling between the cell’s bioactive molecules, such as RNAs, cytokines, and growth factors. MM-derived exosomes experience a higher cell internalization rate than those from MGUS and healthy individuals. It has been found that Heparin sulfate proteoglycans (HSPGs) on the cell surface mediate exosome docking and processing by the cell. Thus, the binding kinetics of MM-derived exosomes and Heparin are studied using an SPR chip functionalized with Heparin-Heparin. SPR sensogram data showed that MM exosomes are more bound to Heparin than other exosomes (healthy and MGUS). High-affinity binding with a dissociation constant K_d_ of 0.88 nM was observed [142].

The challenge of tumor heterogeneity profiling due to variations in molecular composition underscores the need to characterize EVs based on multiple biomarkers. SPR’s multiplexing features come to the fore, offering a promising solution. The molecular profiling of exosomes derived from ovarian cancer cell conditioned media on a nano-plasmonic exosome (nPLEX) chip has revealed high expression of CD24 and EpCAM. The periodic nanoholes patterned in a metal film, which produce high sensitivity for SPR signals, are a key component of this process. Each nanohole, acting as an individual testing chamber, is functionalized with monoclonal antibodies. The binding of exosomes to the ligand is then detected using SPR spectroscopy [143].

As previously mentioned, blood contains a higher concentration of EVs (10^8^–10^11^ EV/mL), making blood-derived EVs suitable for minimally invasive techniques. The molecular profile of exosomes derived from MCF-7 cells and whole blood reveals similarities in biomarkers such as tetraspanins (CD), EpCAM, and HER2 [144]. Before analysis, it is essential to enrich EVs from body fluids. Directly detecting EVs from biological samples without prior enrichment and subsequent label-free molecular profiling is a critical step in liquid biopsy. Sina, A. A. I. et al. [145] successfully detected exosomes using CD9 and CD63 and performed molecular profiling of immunoaffinity-captured bulk exosomes from the serum of breast cancer patients, identifying the presence of the HER2 growth factor. In this process, exosomes are captured using the exosome-specific biomarkers CD9 and CD63, and then tested for the presence of the breast cancer biomarker HER2 using a HER2 antibody.

The SPR sensing method requires a simple setup for detection. Efficient utilization of SPR would make this technology a point-of-care device. An intensity-modulated, compact SPR biosensor (25 × 10 × 25 cm) developed by Liu, C et al. [146] showed that an SPR can detect Exosomal protein originating from lung cancer (A549) exosomes. The detection setup includes a laser source, an optical splitter, prisms, and photodetectors. The optical splitter splits the incoming laser beam into two streams of equal intensities. One stream is directed towards the first photodetector, thus making it a reference beam. The second stream interacts with biomolecule samples that are biotinylated antibodies of anti-EGFR, anti-PD-L1, and anti-IgG, and the subsequent intensities are recorded. The device offers a sensitivity of 9.258 × 10^3^%/RIU and a resolution of 8.311 × 10^−6^ RIU.

Su, J. et al. [147] developed an optical resonator that detects changes in the effective index of refraction. A frequency-locking optical whispering evanescent resonator (FLOWER) consists of a light source and a glass microtoroid, as shown in Figure 8. Light is evanescently coupled into a glass microtoroid optical resonator using an optical fiber. This light internally reflects inside the rim of the microtoroid, generating an evanescent field. At the resonance frequency, light constructively interferes, causing light recirculation and signal amplification. FLOWER detects changes in the effective index of refraction (defined as the ratio of the index of refraction of the particle to the index of refraction of its surrounding media) of the microtoroid as particles enter its evanescent field. These index of refraction changes are detected by monitoring corresponding changes in the resonance frequency of the microtoroid. Mouse serum-derived exosomes were tested for tumor progression using anti-CD81.

SPR can be used as a tool to study both quantitative and qualitative data. The reaction between biomarker (CD63) and antibody (anti-CD63) can be tracked over time by measuring SPR signals over time [148]. The low sample requirement and exceptional sensitivity (1.02–1.04 × 10^6^ RU/RIU) of surface plasmon resonance (SPR) make it one of the superior tools for analyzing biomolecules present in trace amounts. Thus, SPR is a perfect fit for studying EVs at the early stages of cancer, where circulating markers are in trace amounts.

### 4.4. Surface Plasmon Resonance Imaging

Surface plasmon resonance imaging (SPRi) is an improved version of SPR aimed at high-throughput analysis (up to 1000 interactions) [149]. The light source for SPRi is coherent and polarized, instead of the polychromatic light in SPR. This enables SPRi to cover a broader range of sensing areas. Like SPR, SPRi measures the change in refractive index in the local dielectric medium and is translated into images using a charge-coupled device (CCD) that detects reflectivity, as shown in Figure 9. The high-resolution CCD camera provides real-time images across the array format with up to hundreds of active spots. SPRi distinguishes itself from SPR by the angle of incident light and the wavelength used, which remain constant in the process. Hence, changes in reflected light intensity are proportional to any variation in the refractive index near the metal surface. Surface plasmon resonance imaging-based biosensors for multiplex and ultrasensitive detection of exosomes use microarrays consisting of different antibodies, as shown in Figure 9 [150]. SPRi faces challenges in its application in exosome analysis due to the heterogeneity of exosome biomolecules.

### 4.5. Localized Surface Plasmon Resonance (LSPR)

LSPR is the latest detection method amongst the plasmonic sensing platforms. Surface plasmons generated via the interaction of incident photons with nanoparticles are localized (Figure 10). Such evanescent fields are a few nanometers in size (up to a hundred nanometers) from the metal surface. Such a small field only detects biomolecules bound to metal surfaces and avoids bulk sensing. LSPR offers to develop rapid, label-free detection strategies. Adaptation of LSPR in biosensing is relatively rapid owing to the high sensitivity of the Atto molar (10^−18^) concentration [151].

Similar to Surface Plasmon Resonance (SPR), plasmons are generated by the interaction of plasmonic particles (such as nanoparticles) with incident light. However, when the particles are small, the plasmon does not propagate like a wave across the surface. Instead, it is confined to the surface of the nanoparticle, creating a localized electromagnetic field. This phenomenon, where the plasmon is confined to the nanoparticle surface, is known as Localized Surface Plasmon Resonance (LSPR). This LSPR field is generated through the resonant oscillation of free electrons in the conduction band of the material. The resonance frequency of these oscillations is dependent on the wavelength of the incident light. When the incident light wavelength matches the resonant frequency of the nanoparticle, enhanced absorption, scattering, and local electromagnetic fields occur at the nanoparticle surface. These increases in absorption and scattering are exploited in LSPR biosensing applications, where changes in resonance are used to detect interactions such as binding events or changes in the local environment.

The LSPR response of a plasmonic material can be tuned with the knowledge of Mie theory. Mie theory provides an analytical solution to Maxwell’s equation. For gold nanoparticles, often used in LSPR sensing applications, spherical boundary conditions are used to characterize the extinction spectra given by Equation (3) [152]:(3)$$Q_{ext}=\frac{2}{x^{2}}\sum (2n+1)(a_{n}+b_{n})$$
where *Q_ext_* is a dimensionless quantity representing the total loss of light intensity due to both scattering and absorption by a particle.$$Q_{ext}=\frac{\sigma_{ext}}{\pi r^{2}}$$

*σ_ext_* is the extinction cross-section is the effective area over which the particle absorbs and scatters light. *r* is the radius of the spherical particle.(4)$$Q_{ext}=Q_{abs}+Q_{scat}$$

*Q_abs_* is the absorption efficiency, which quantifies the fraction of light that is absorbed by the particle. *Q_scat_* is a scattering efficiency that quantifies the fraction of light scattered by the particle.

*x* is size parameter, given by (2π*r*/λ). λ wavelength of incident light.

*n* represents the multipole order, corresponding to the different orders of the interaction (dipole, quadrupole, etc.).

*a_n_* and *b_n_* are the Mie coefficients that depend on the size parameter.

The wavelength shift due to analyte binding to nanoparticles is given by Equation (5) [153].(5)$$∆\lambda =m(∆\eta )[1-exp(\frac{-2d}{l_{d}})]$$
where Δλ is wavelength shift, m is the is the refractive index sensitivity, Δη is change in refractive index due to analyte binding, and *l*_d_ is the electromagnetic field decay length.

Thus, in an LSPR sensor, the size of the nanoparticle, refractive index sensitivity, and electromagnetic field decay length all can affect the electromagnetic field.

AuNPs have been widely used in exosome research owing to their biocompatibility as well as excellent plasmonic properties. The relative bindings of exosomes, AuNPs, and Aptamers have been strategically used to differentiate protein concentration across exosomes originating from different cell lines. The principle of analysis is that, in the presence of high salt media, AuNP aptamers conjugate when introduced with a non-specific exosome; the non-specific weaker bond between AuNP and the aptamer is broken, and a stronger specific bond between the aptamer and exosomes is formed, thus displacing AuNPs from the aptamer. Without aptamers attached to AuNPs, they form aggregates in high-salt media. Hence, this method predicts the specificity and bonding of aptamer and exosomes by analyzing the AuNP aggregate in the solution, which can be studied using absorption spectroscopy. AuNPs have blue and red shifts during their dispersion to the aggregation state change. The AuNP colorimetric study is an approach where the interaction of protein molecules can be studied. As a proof of concept, this method was used to study the relative concentration of CD63 surface protein across four different lines: HeLa, PC3, Ramos, and CEM. This study showed that exosomes from Ramos and CEM could not produce significant aggregation of AuNPs; hence, there was less bonding between the aptamer and exosomes. HeLa and PC3 samples had more aggregates of AuNPs, noting a higher presence of CD63 in the exosomes derived from these two cell lines. This study was further extended to different surface molecules, such as PTK7 (Protein kinase-7), EpCAM (Epithelial cell adhesion molecule), PDGF (platelet-derived growth factor), and PSMA (prostate-specific membrane antigen), across these cell lines. This study could distinguish subtle changes in the concentration of surface molecules. The results agree with previous findings from other types of studies [154].

Recently, our group developed a liquid biopsy chip for breast cancer diagnosis that integrates label-free techniques into EV analysis in a microfluidic platform. The device comprises a mixing chamber, a detection chamber, and a sedimentation chamber. MCF-7 cell culture media is injected into the microfluidic channel and mixed with Vn96 functionalized magnetic nanoparticles, as shown in Figure 11. Immunoaffinity-captured EVs are magnetically isolated. Proteinase K treatment on magnetically isolated EVs is performed for downstream processing. Protein quantification performed through ddPCR shows excellent gene copies in captured EVs up to 50× dilution, indicating the effectiveness of the microfluidic chip in isolating EVs for liquid biopsy [155].

Detection of extracellular vesicles (EVs) in microfluidic platforms utilizing plasmonic techniques holds considerable promise for liquid biopsy applications, particularly in non-invasive cancer diagnostics. EVs, including exosomes and microvesicles, transport molecular markers such as proteins, RNA, and lipids from their cells of origin, offering valuable insights into disease states. Plasmonic techniques, such as Surface-Enhanced Raman Spectroscopy (SERS) and localized surface plasmon resonance (LSPR), significantly enhance the sensitivity and specificity of detecting these vesicles in complex biological samples such as cell culture, serum, etc. These techniques are integrated with miniaturized devices in microfluidic platforms, enabling efficient capture and analysis of EVs from small sample volumes.

Table 5 lists the microfluidic devices integrated with label-free plasmonic EV detection techniques using EV markers. Common markers used for EV characterization in liquid biopsy include cancer-related proteins, such as epidermal growth factor receptor (EGFR) [146], HER2 [145], tetraspanins [143,156], and EV markers HSP90 [155], as well as biomarkers associated with tumor progression and metastasis [142].

Overall, plasmonic techniques are better suited for clinical applications than fluorescent techniques. SERS enables real-time monitoring, with high sensitivity for quick detection of molecular signatures in clinical samples without lengthy preparatory steps. SPR also allows real-time tracking of biomolecule interactions, such as EVs binding to sensor surfaces, providing immediate feedback without additional processing or washing. Since SPR detects changes in the refractive index as analytes bind, it allows continuous monitoring of interactions with exceptionally high sensitivity. In addition, the degree of refractive index changes correlates directly with the amount of biomolecule bound, thus offering high quantitative accuracy.

In contrast, while fluorescence can be detected in real-time, it requires sample preparation and the addition of dyes, which can introduce variability and delay. Both are critical in clinical settings. Furthermore, fluorescence-based assays can face challenges in quantification due to variability in labelling efficiency, differences in probe binding affinity, and non-linearities in fluorescence emission at different concentrations. Fluorescence also suffers from quenching effects in certain biological environments, which can complicate quantification. Furthermore, signals can suffer from photobleaching over time and be affected by background fluorescence from other sample components, which reduces accuracy.

The biocompatibility of plasmonic techniques makes them suitable for liquid biopsy applications. SERS effectively identifies low concentrations of biomarkers in various types of clinical samples, such as blood, urine, and human tissue. SPR is robust in complex biological environments. Fluorescence can be hindered by autofluorescence and requires more purification, making it less practical in clinical settings.

While the initial setup cost for SPR and SERS equipment can be high, their operational costs are often lower than those of fluorescence-based systems, as no labels or dyes are needed. SERS can utilize inexpensive gold or silver nanoparticles, and Raman spectroscopy is increasingly available in clinical settings. In contrast, fluorescence detection requires costly fluorescent dyes and advanced imaging systems, adding to both cost and complexity, especially when analyzing many samples in clinical environments.

A plasmonic biosensor utilizing a tapered optical fiber (TOF) has been developed for the sensitive identification of various microRNAs linked to prostate cancer in human serum. This sensor effectively differentiated between cancerous and noncancerous samples with detection limits ranging from 179 to 580 Atto moles. The TOF plasmonic biosensor shows potential as a point-of-care diagnostic tool for cancer detection in a clinical setting [157].

The fusion of microfluidic handling and plasmonic detection creates a highly sensitive, real-time method for isolating and analyzing EVs. It is a powerful tool for early cancer diagnosis, prognosis, and treatment response monitoring. With detection limits as low as picograms per milliliter, microfluidic platforms can identify analytes in trace amounts. Therefore, a microfluidic platform equipped with plasmonic techniques is particularly well-suited for EV-based liquid biopsy applications.

## 5. Research Outlook and Conclusions

Liquid biopsy has the potential to offer a non-invasive diagnosis of cancer at an early stage. Sample collection, sample volume requirement, and ease of testing make liquid biopsy an attractive and reliable diagnostic technique for early-stage cancer. Successful integration of liquid biopsy into the ‘cancer diagnostic eco-system’ depends on its predictive capabilities. EVs are apt analytes for liquid biopsy owing to their availability in blood in early-stage cancer, their half-life in blood, and their ability to cross the blood–brain barrier. Conventionally handling EVs is a complex task involving several steps of enrichment. Microfluidic devices are excellent, tiny devices capable of handling EVs for testing. Plasmonic and microfluidics pave the way to developing liquid biopsy and point-of-care diagnosis. Microfluidic platforms promise effective ways to isolate EVs from complex biosamples such as blood.

Nano-plasmonic techniques are excellent tools to analyze EVs for oncogenes. Raman and SERS are superior to other techniques for chemical fingerprints, while SPR and LSPR are excellent techniques with outstanding sensitivity and limited detection. EV-based liquid biopsy is the advancement of technology towards non-invasive/minimally invasive cancer diagnosis, thereby reducing the physical burden for patients who are already dealing with life-threatening health conditions.

The translation of extracellular vesicle (EV)-based liquid biopsy technologies utilizing plasmonic techniques to clinical practice faces several challenges. One of the key hurdles is the lack of standardized methods for isolating EVs from blood and other biological fluids, which is crucial for ensuring reproducible results. Current isolation techniques vary widely in terms of their efficiency, specificity, and scalability, leading to inconsistent EV populations and, consequently, unreliable biomarker detection.

Standardization of EV isolation is necessary to avoid misinterpretation of the data that might arise from heterogeneity among extracellular vesicles (EVs). The varying size and content of the EV subtypes make it challenging to identify and analyze them uniformly. Thus, different studies that focus on different subsets of EVs lead to inconsistent findings. EVs’ biological cargo varies widely depending on the originating cell type, the physiological state of the cell (e.g., cancerous vs. normal), and external factors like hypoxia or stress.

When an assay targets a specific EV subtype, it might miss other biologically significant EVs that lack the targeted biomarker, skewing results. For example, detection methods based on specific surface proteins (e.g., CD63 for exosomes) will overlook microvesicles or apoptotic bodies that lack the same markers. Thus, researchers must account for this variability when designing studies, interpreting data, and developing EV isolation, detection, and analysis technologies. To overcome these challenges, more precise and reproducible isolation techniques are needed. Furthermore, an advanced method for characterizing an EV subtype’s molecular and functional diversity is necessary. Addressing these issues is crucial for realizing the full potential of EVs as biomarkers for cancer diagnosis, prognosis, and therapy.

Scalability is another major challenge when integrating plasmonic technologies with EV-based liquid biopsy for clinical use. While laboratory-scale experiments have shown the potential of plasmonic sensors in detecting EVs, translating these techniques to high-throughput clinical environments presents significant logistical and technical difficulties. A key concern in scalability is the miniaturization of equipment, which must maintain the high precision and sensitivity for EV isolation while being cost-effective. Advanced manufacturing techniques, such as 3D printing, could help overcome these challenges by enabling mass production, thus reducing costs and improving scalability. However, the production and functionalization of plasmonic nanoparticles and microfluidic chips must be optimized for large-scale use, ensuring consistency and high-quality standards. Furthermore, cost control remains a critical barrier; scaling the technology for clinical applications will require careful management of production costs to make it affordable for widespread adoption. Addressing these technological hurdles is essential for realizing the full potential of these technologies in real-world, clinical applications.

Furthermore, regulatory approval for such innovative technologies is a complex process, as medical devices must demonstrate both safety and efficacy in clinical trials. The regulatory pathway for plasmonic-based liquid biopsy technologies is still in its infancy, and navigating the approval process can be slow and uncertain, particularly given the novelty of the approach and the potential risks associated with introducing new nanomaterials into clinical practice. This regulatory uncertainty and the challenges of scaling and standardizing EV isolation methods pose substantial barriers to the widespread clinical adoption of EV-based liquid biopsies using plasmonic technologies.

While EV-based liquid biopsy technologies hold significant promise, alternative techniques, such as Polymerase Chain Reaction (PCR) and single-cell sequencing, are also gaining traction. PCR is renowned for its high sensitivity in detecting specific genetic markers, making it a formidable tool in cancer detection [158]. Nonetheless, its reliance on targeted assays limits its capacity to identify a wide array of biomarkers within complex samples. In contrast, single-cell sequencing has emerged as a groundbreaking methodology, enabling the analysis of individual cells to reveal rare mutations and unique gene expression profiles. While it provides comprehensive insights, the high cost and challenges of scaling this technology for large clinical applications persist.

Each EV-based technology has the potential to be integrated into early-stage cancer detection owing to its unique strengths. Single-cell sequencing offers an in-depth understanding of cellular heterogeneity but faces hurdles in scalability and cost-effectiveness for clinical use. PCR, while affordable and widely available, is constrained by its inability to simultaneously analyze complex panels of biomarkers. Conversely, EV-based liquid biopsy presents a less invasive option with the potential to analyze multiple biomarkers across various stages of cancer; however, it still contends with significant challenges related to scalability, cost control, and standardization.

## Figures and Tables

**Figure 1 ijms-26-06352-f001:**
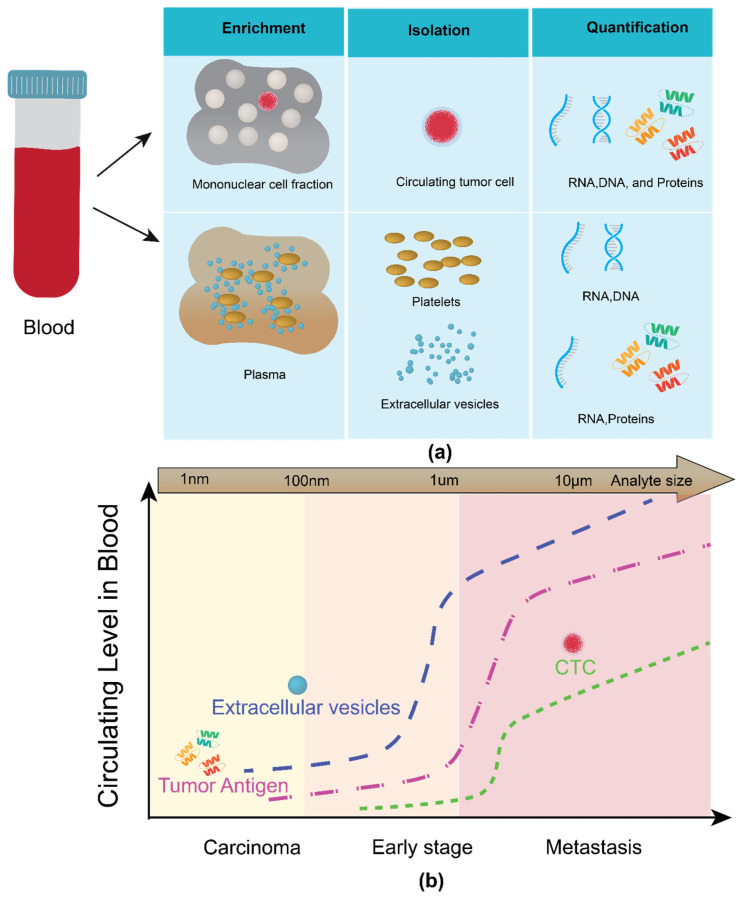
(**a**) Liquid biopsy analytes in blood, (**b**) relative levels of liquid biopsy analytes circulating in blood. Modified from He M., Zeng Y. [5]. Copyright 2016, J Lab Automation.

**Figure 2 ijms-26-06352-f002:**
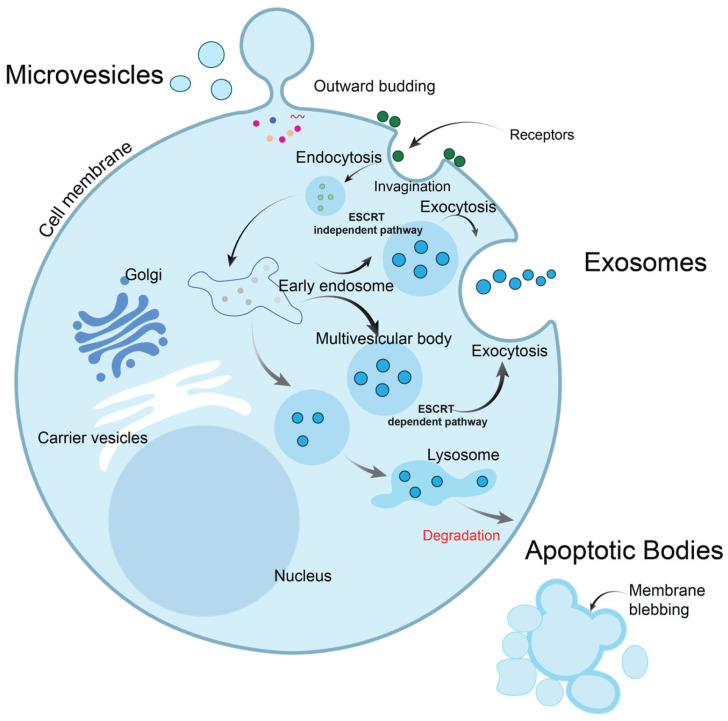
Biogenesis of extracellular vesicle (EV) subtypes. Exosomes are formed via the Endosomal Sorting Complexes Required for Transport (ESCRT) pathway. ESCRT proteins aid in organizing cargo into the intraluminal vesicles (ILVs) found within multivesicular bodies (MVBs), which eventually develop into exosomes. Exosomes can also be formed through ESCRT-independent pathways.

**Figure 3 ijms-26-06352-f003:**
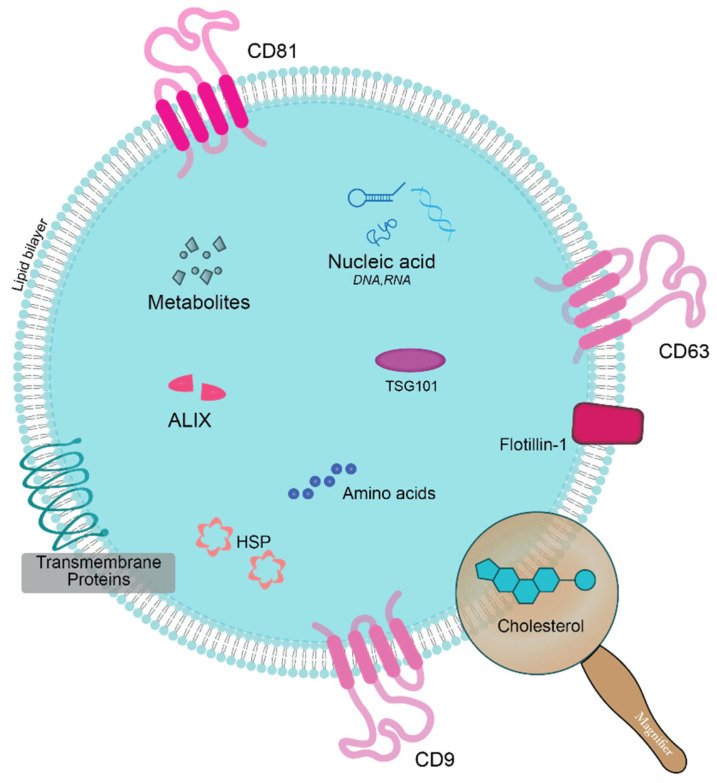
Major biomolecules that are contained in exosomes.

**Figure 4 ijms-26-06352-f004:**
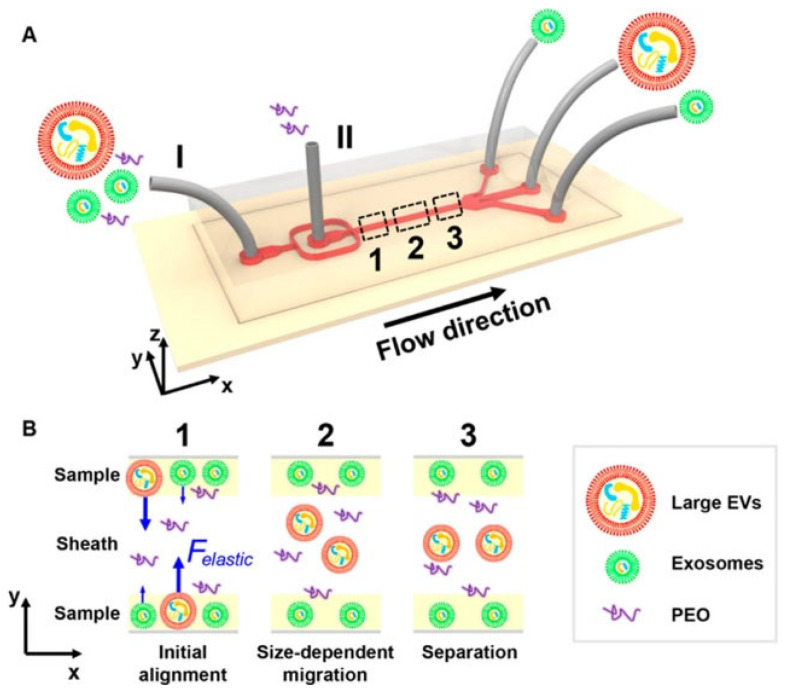
(**A**) Schematic of the microfluidic chip for exosome separation from large EVs, (**B**) Illustration of separation mechanism in viscoelastic microfluidics [104]. Reprinted with permission from Liu, C., Guo, J., Tian, F., Yang, N., Yan, F., Ding, Y., Wei, J., Hu, G., Nie, G. and Sun, J., 2017. Field-free isolation of exosomes from extracellular vesicles by microfluidic viscoelastic flows. ACS nano, 11(7), pp.6968-6976. https://doi.org/10.1021/acsnano.7b02277. Copyright 2017 American Chemical Society.

**Figure 5 ijms-26-06352-f005:**
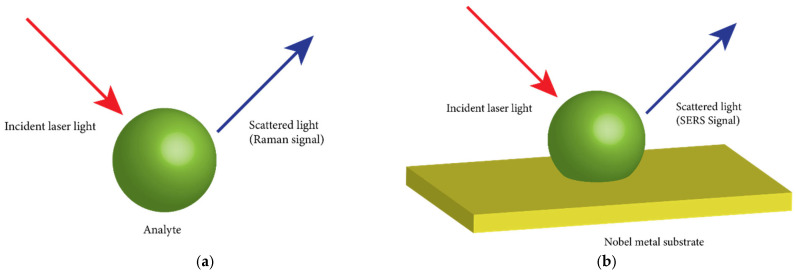
Schematic of (**a**) Raman scattering; (**b**) surface-enhanced Raman scattering.

**Figure 6 ijms-26-06352-f006:**
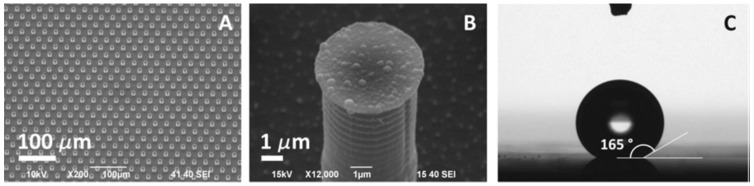
A periodic hexagonal pattern of cylindrical pillars yielding a superhydrophobic surface (**A**); one silicon micropillar tailored with a randomly distributed silver nanograins assembly for superior SERS analysis (**B**); a drop, positioned upon the surface as in (**A**), experiencing a contact angle as large as 165 (**C**) [125]. Reprinted with permission from Tirinato, L., Gentile, F., Di Mascolo, D., Coluccio, M.L., Das, G., Liberale, C., Pullano, S.A., Perozziello, G., Francardi, M., Accardo, A. and De Angelis, F., 2012. SERS analysis on exosomes using super-hydrophobic surfaces. Microelectronic Engineering, 97, pp.337–340. https://doi.org/10.1016/j.mee.2012.03.022. Copyright 2012 Elsevier B.V.

**Figure 7 ijms-26-06352-f007:**
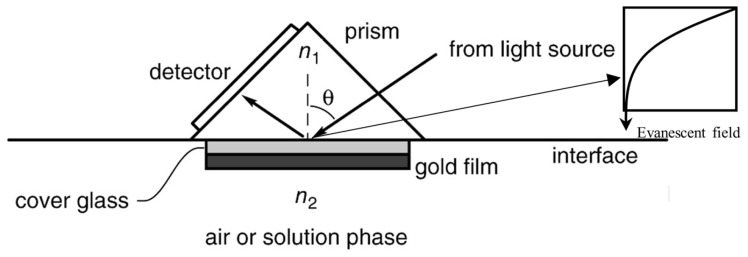
Schematic of surface plasmon resonance [140]. Reprinted with permission from Tang, Y., Zeng, X. and Liang, J., 2010. Surface plasmon resonance: an introduction to a surface spectroscopy technique. Journal of chemical education, 87(7), pp. 742–746. https://doi.org/10.1021/ed100186y. Copyright 2010 American Chemical Society.

**Figure 8 ijms-26-06352-f008:**
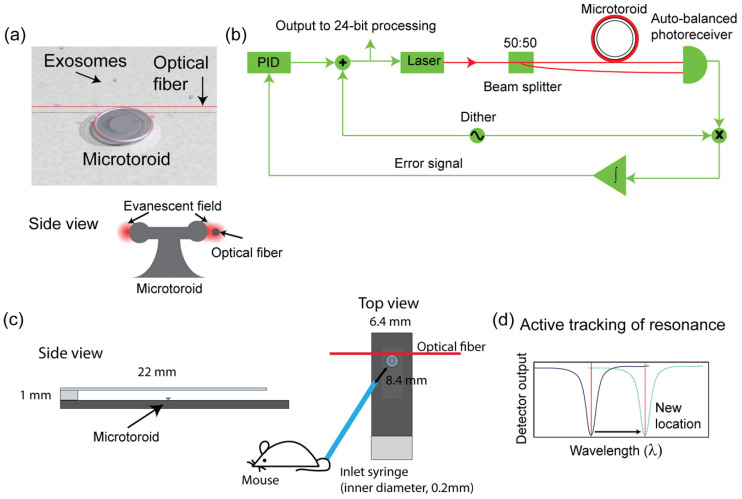
Frequency locking optical whispering evanescent resonator (FLOWER). (**a**) A microtoroid where light is coupled into it through an optical fiber via evanescent coupling. (**b**) Block diagram of the sensing control system. (**c**) Schematic of the sample cell from the side and top view. (**d**) Schematic of resonance frequency change [147]. Reprinted with permission from Su, J., 2015. Label-free single exosome detection using frequency-locked microtoroid optical resonators. Acs Photonics, 2(9), pp. 1241–1245. https://doi.org/10.1021/acsphotonics.5b00142. Copyright 2015 American Chemical Society.

**Figure 9 ijms-26-06352-f009:**
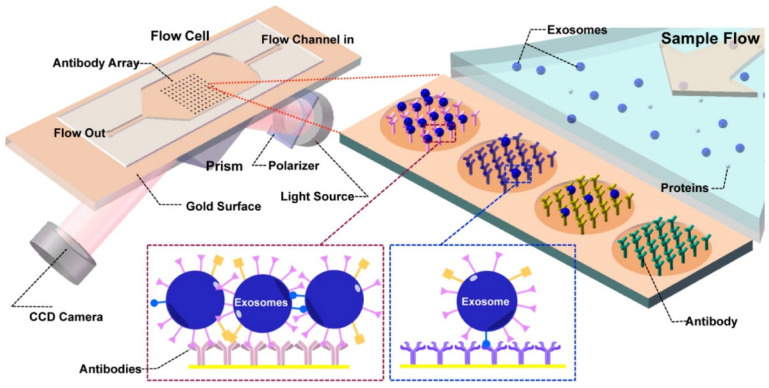
Schematic view of SPRi in combination with antibody microarray to capture and detect exosomes in cell culture supernatant. Exosomes binding onto antibodies (inset) [150]. Zhu, L., Wang, K., Cui, J., Liu, H., Bu, X., Ma, H., Wang, W., Gong, H., Lausted, C., Hood, L. and Yang, G., 2014. Label-free quantitative detection of tumor-derived exosomes through surface plasmon resonance imaging. Analytical chemistry, 86(17), pp. 8857–8864. https://doi.org/10.1021/ac5023056. Licensed ACS Author Choice License, which permits redistribution of the article or any adaptations for non-commercial purposes.

**Figure 10 ijms-26-06352-f010:**
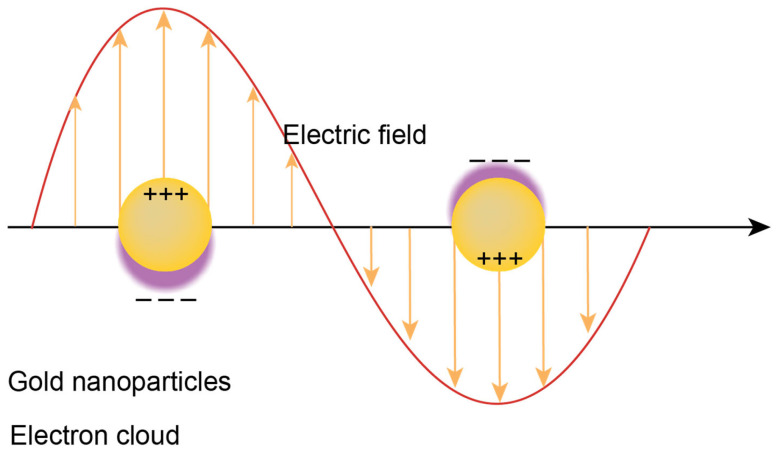
Schematic of localized surface plasmon field.

**Figure 11 ijms-26-06352-f011:**
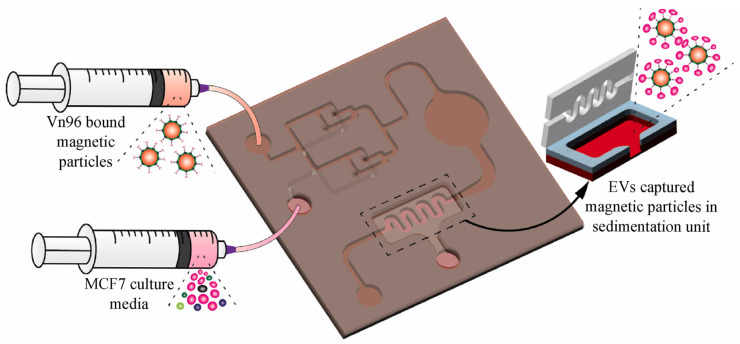
Schematic of EV-isolation from the CCM when Vn96-bound magnetic particles and CCM were infused into the device [155]. Reprinted with permission from Bathini, S., Pakkiriswami, S., Ouellette, R.J., Ghosh, A. and Packirisamy, M., 2021. Magnetic particle based liquid biopsy chip for isolation of extracellular vesicles and characterization by gene amplification. Biosensors and Bioelectronics, 194, p. 113585. https://doi.org/10.1016/j.bios.2021.113585. Copyright 2021^©^ 2021 Elsevier B.V.

**Table 1 ijms-26-06352-t001:** List of approved liquid biopsy tests in North America: SC: Screening, D: Diagnostic, PT: Post-treatment.

Tests (Company) ☐SC|☐D|☐PT	Cancers	Marker	Sample	Comments	Refs.
Guardant360 (Guardant Health, Palo Alto, CA, USA) ☑D	Lung (NSCLC), Breast, Colorectal, Prostate	100 + ctDNA	Blood	Effective for advanced stage (stage III or IV) cancer. FDA approved in 2020.	[8,9,10]
FoundationOne Liquid (Foundation Medicine, Boston, MA, USA) ☑D	NSCLC	70 + ctDNA	Blood	For patients with advanced cancer. FDA approved in 2020.	[11]
Epi pro Colon (Epigenomics AG Heidelberg, Germany) ☑SC, ☑D	colorectal cancer	methylated Septin 9 DNA	Blood	For individuals aged 50 and older who are at average risk for colorectal cancer, this is the first blood-based test available. FDA approved in 2016.	[12]
Cologuard (Exact Sciences, Madison, WI, USA) ☑D	colorectal cancer	DNA	Stool	The first non-invasive DNA screening test for colorectal cancer is intended to screen adults 45 years of age and older who are at average risk for colorectal cancer. FDA approved in 2014.	[13]
CELLSEARCH CTC kit (Menarini Silicon Biosystems, Inc., Huntingdon Valley, PA, USA) ☑D	Breast, Prostate and Colorectal	CTC	Blood	Test approved by Health Canada in 2010. First FDA-cleared test (2004) for the enumeration of circulating tumor cells in peripheral blood.	[14]
Laboratory-developed test (LDT) regulated by Clinical Laboratory Improvement Amendments (CLIA)					
Galleri (GRAIL, Menlo Park, CA, USA) ☑SC	Multiple cancer screening	ctDNA (methylation markers and machine learning for cancer detection)	Blood		[15]
OncoBEAM (Sysmex Inostics, Baltimore, MD, USA) ☑SC, ☑PT	Lung, Colorectal, and other	ctDNA	Blood		[16,17]
CancerSEEK (Johns Hopkins University, Baltimore, MD, USA) ☑SC	Ovarian, Liver, Stomach, and other.	ctDNA	Blood	ctDNA and protein markers for early cancer detection.	[18]
Pathfinder (Freenome, Brisbane, CA, USA) ☑SC	Colorectal cancer	ctDNA	Blood	a combination of DNA, RNA, and protein biomarkers.	[19]

**Table 3 ijms-26-06352-t003:** Exosomal markers across different cancer types.

Cancer Type	Exosomal Biomarker	Biomarker Type	Biofluid	Indication	Clinical Sample Size	Refs.
Lung cancer	miR-222-3p	miRNA	Serum	Prognosis	TP N = 50	[56]
	EGFR T790M	mRNA	Plasma		TP N = 84 [57]	[57,58]
	miR-181-5p, miR-30a-3p, miR- Adenocarcinoma-specific: 30e-3p, and miR-361-5p were SCC specific: miR-10b-5p, miR-15b-5p, and miR-320b	miRNA	Plasma		TP N 46, HC N = 42, S = 60	[59]
	miR-193a-3p, miR-210-3p and miR-5100	miRNA	Bone marrow and Plasma	Diagnosis and prognosis	TP N = 41, HC N = 30	[60]
	circSATB2	Circular RNA	Cell line H460, A549 and H1299			[61]
	NY-ESO-1	Antigen	Plasma			[62]
	TTF-1 and miR-21	Protein	Serum	Diagnosis	NA	[63]
Breast cancer	miR-1246	miRNA	Plasma	Diagnosis	TP N = 46, HC N = 28	[64]
	miR-21, miR-105 and miR-222	miRNA	Serum	Diagnosis	TP N = 53, HC N = 8	[65]
	Glycoprotein (MUC1)	Protein	Cell line MCF7&MDA-MB-231			[66]
	PKG1, RALGAPA2, NFX1, TJP2	Protein	Plasma			[67]
	HER2	Protein	Plasma			[68]
	CD82	Protein	Serum and plasma	Diagnosis	TP N = 80, BTP N = 80, HC N = 80	[69]
	miR-375	miRNA	Serum	Diagnosis	TP N = 17, HC N = 12	[70]
Gastric cancer	circSHKBP1	Circular RNA	Serum	Diagnosis	TP N = 20, HC N = 20	[71]
	HOTTIP	long non-coding RNA	Serum	Diagnosis and prognosis	TP N = 126, HC N = 120	[72]
Early-stage GC	lncUEGC1	long non-coding RNA	Plasma	Diagnosis	TP N = 10, HC N = 5	[73]
Rectal cancer	miR-30d-5p, miR-181a-5p and miR-486-5p	miRNA	Plasma	Diagnosis and prognosis	TP N = 24, HC N = 5	[74]
HCC	tRNA-ValTAC-3, tRNAGlyTCC-5, tRNA-ValAAC-5, and tRNA-GluCTC-5	miRNA	Plasma	Diagnosis	TP N = 35, HC N = 11	[75]
	circUHRF1	Circular RNA	Plasma	Diagnosis	TP N = 240, HC N = 20	[76]
Early-stage HCC	miR-21 and miR-10b	miRNA	Serum	Prognosis	TP N = 124	[77]
Pancreatic cancer	KRAS	mRNA	Plasma	Diagnosis and prognosis	TP N = 127, HC N = 136	[78,79]
	CKAP4	Protein	Serum	Diagnosis	TP N = 47, HC N = 18	[80]
	Glypican 1(GPC1), Migration inhibition factor (MIF)	Protein	Serum		TP N = 71, HC N = 32	[81]
Prostate cancer	AR-V7	Androgen receptor	Plasma	Prognosis	TP N = 36	[82]
	miR-196a-5p and miR-501-3p	miRNA	Urine	Diagnosis	TP N = 48, HC N = 28	[83]
	miR-1290 and miR-375	miRNA	Plasma		TP N = 23, HC N = 50	[84]
	PSA	Antigen	Plasma			[85]
	GGT1	Gene	Serum			[86]
	PTENP1	Gene	Plasma	Diagnosis	TP N = 50, HC N = 60	[87]
Bladder cancer	lncLNMAT2	long non-coding RNA	Serum and urine	Diagnosis and prognosis	TP N = 206, HC N = 120	[88]
	lncRNAs(SPRY4-IT1, MALAT1 and PCAT-1)	long non-coding RNA	Urine	Diagnosis and Prognosis	TP N = 184, HC N = 184	[89]
Colorectal	Glypican-1(miR-96-5p and miR-149)	miRNA	Plasma			[90]
	CEA	Antigen	Serum			[91]
Cholangiocarcinoma	AMPN, VNN1, PIGR	Gene	Serum			[92]
Ovarian cancer	E-cadherin	Protein	Ascites	Diagnosis and prognosis	TP N = 35, HC N = 6	[93]
	miR-200b and miR-200c	miRNA	Serum	Diagnosis and prognosis	TP N = 163, BTP N = 20, HC N = 32	[94]
	CD24, EpCAM, CA-125	Protein	Plasma			[95]
Cervical cancer	let-7d-3p and miR-30d-5p	miRNA	Plasma	Diagnosis	NA	[96]
Multiple Melanoma	Ig-BCR	Cell receptor	Serum	Diagnosis	Serum of 5T33MM engrafted mice	[97]
	let-7b and miR-18a	miRNA	Serum	Prognosis	TP N = 156, HC N = 5	[98]
	PMSA3 and lncPMSA3-AS1	Gene	Serum	Prognosis	Bortezomib resistance N = 12, bortezomib sensitivity N = 45	[99]
Melanoma	PD-L1	Protein	Plasma	Diagnosis and prognosis	TP N = 44, HC N = 11	[100]
	BRAFV600E	Gene	Plasma	Prognosis	TP N = 12, HC N = 12	[101]
Glioblastoma	EGFR vIII	mRNA	Serum	Diagnostic		[102]

Abbreviations: TP N: Number of total patients; HC: Number of healthy controls; NA: Data not available.

**Table 4 ijms-26-06352-t004:** Optical and non-optical techniques for EV detection.

Optical Methods	Non-Optical Methods
Optical microscopy	Scanning electron microscope (SEM)
Dynamic Light Scattering (DLS)	Transmission Electron Microscopy (TEM)
Nano Tracking Analysis (NTA)	Atomic force microscopy (AFM)
Fluorescence microscopy	Impedance flow cytometry
Surface Plasmon Resonance (SPR)	Tunable Resistance Pulse Sensing (TRPS)
Localized Surface Plasmon Resonance (LSPR)	Mass spectroscopy

**Table 5 ijms-26-06352-t005:** Various cancer biomarkers found using the plasmonic technique in microfluidic platform.

Biomarker	Cancer Type (Cell Line)	Sample	Detection Technique (Target Molecule)	Statistics	Refs.
CD63	Glioma (U251)	Cell culture	SPR (Antibody)	4.23 × 10^−3^ μg/mL	[156]
CD24 EpCAM	Ovarian (CaOV3)	Cell culture	SPR (Antibody)	1000+ sites for inspection	[143]
EGFR PD-L1	Non-small cell lung cancer (A549)	Cell culture, Serum	SPR (Antibody)	sensitivity of 9.258 × 103%/RIU and resolution of 8.311 × 10^−6^ RIU.	[146]
HER2	Breast cancer (BT474)	Cell culture, Serum	(SPR) (Antibody)	LOD: 2070 Exosomes/μL	[145]
HSPG	Multiple myeloma	Serum	SPR (Protein)	LOD: 5 ng/mL	[142]
HSP90	Breast cancer (MCF7)	Cell culture	LSPR (Vn96)	No data	[155]

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
