# Peer review of "Microfluidic Liquid Biopsy Minimally Invasive Cancer Diagnosis by Nano-Plasmonic Label-Free Detection of Extracellular Vesicles: Review"

_ijms, 2025, doi:10.3390/ijms26136352_

Round 1

Reviewer 1 Report

Comments and Suggestions for Authors

The review is well written and comprehensive in its content. The tables and figures are generally well structured and clear. Although the manuscript is quite long, I do not find it excessively verbose.

A few minor comments:
AF4 should also be mentioned among the vesicle isolation techniques:

https://doi.org/10.1016/j.chroma.2020.461773

DOI: 10.3390/s23239432

The text in green in Figure 2 is hard to read. Furthermore, the distinction between early and late endosomes should be discussed:

Trends in cell biology 25.6 (2015): 364-372.

Reviewer 2 Report

Comments and Suggestions for Authors

In this review, the authors explored the application of label-free detection methods based on microfluidics and nanoplasma technology in liquid biopsy, focusing on extracellular vesicles (EVs) as biomarkers for early diagnosis of cancer. The specific introduction includes the challenges and opportunities faced by liquid biopsy, the role of extracellular EVs, the application advantages of microfluidics and nanoplasma technology, and the challenges of clinical transformation. The author made a relatively detailed discussion and summary, but there are still some problems in the language logic and organization of the manuscript. Therefore, I suggest that the author consider and modify the following queries before publishing the manuscript in Journal of Molecular Sciences. The main contents include:

  1. The English of the manuscript needs further improvement. For example, on page 14, line 409, the repeated use of plasmonic techniques leads to redundancy; “The membranes has an elevated level of amino phospholipids...”, “has” should be replaced with “have”.
  2. On page 15, line 436, the authors did not explain why optical trapping can solve the background signal problem.
  3. Some references are inconsistent in format, authors should check carefully.
  4. On page 21, line 690, the authors did not provide relevant detection limit data for SPR to support the conclusion: “SPR is a perfect fit for studying EVs at the early stages of cancer, where circulating markers are in traces”.
  5. The authors mentioned the advantages of combining microfluidics with nanoplasma technology, but did not discuss in depth the technical bottlenecks that may be encountered in practical applications, such as the miniaturization of equipment, cost control, and the feasibility of large-scale production. These factors are crucial for the clinical translation of technology.
  6. The authors mainly focus on nanoplasma technology, but do not fully compare and analyze it with other emerging technologies (such as digital PCR, single-cell sequencing, etc.). This comparison can help readers better understand the unique advantages and limitations of nanoplasma technology in liquid biopsy.

Round 2

Reviewer 2 Report

Comments and Suggestions for Authors

The revised manuscript generally addresses the raised comments. My opinion is to accept it for publication.

Author Response

We thank the reviewer for their valuable feedback and constructive suggestions, which have helped improve the quality of our manuscript.